# Differential protein expression profiles in human sperm from teratozoospermic and normozoospermic men identify LTBP1 and TGF-βR1 as potential biomarkers within the TGF-β signalling pathway

Paweena Kaewman[1,2☯], Sutisa Nudmamud-Thanoi[ID][1,2☯*], Patcharada Amatyakul[3,4], Chuchard Punsawad[5,6], Sittiruk Roytrakul[ID][7], Samur Thanoi[8☯*]

1 Department of Anatomy, Faculty of Medical Science, Naresuan University, Phitsanulok, Thailand, 2 Centre of Excellence in Medical Biotechnology, Naresuan University, Phitsanulok, Thailand, 3 Department of Obstetrics and Gynecology, Faculty of Medicine, Naresuan University, Phitsanulok, Thailand, 4 Naresuan Infertility Centre, Faculty of Medicine, Naresuan University, Phitsanulok, Thailand, 5 School of Medicine, Walailak University, Nakhon Si Thammarat, Thailand, 6 Research Centre in Tropical Pathobiology, Walailak University, Nakhon Si Thammarat, Thailand, 7 National Center for Genetic Engineering and Biotechnology, National Science and Technology Development Agency, Pathum Thani, Thailand, 8 School of Medical Sciences, University of Phayao, Phayao, Thailand

☯ These authors contributed equally to this work.
* sutisat@nu.ac.th (SN-T); samur.t@up.ac.th (ST)

## Abstract

Among cases of infertility, up to 50% are attributed to male infertility factors. Male infertility is associated with alterations in sperm proteins that are essential for normal sperm function. Alterations in proteins within the transforming growth factor-beta (TGF-β) signalling pathway are linked with several types of male infertility. However, the underlying mechanisms remain unclear. In this study, we proposed to use proteomic analysis to identify the proteomic profile in human sperm from normozoospermic (NOR) and teratozoospermic (TER) men. The results indicated 39 overlapping proteins associated with the TGF-β signalling pathway. Six proteins that were differentially expressed and had a log2 fold change of ≥ 1 or ≤ −1 were considered to be the differentially expressed proteins in human sperm between the groups. Among these proteins, the latent-transforming growth factor beta-binding protein 1 (LTBP1) was significantly increased in the TER group compared to the NOR group. Immunocytochemistry revealed that the protein expression of TGF-β receptor type 1 (TGF-βR1) was localised in the human sperm head. It was also significantly increased in the TER group. Validation analysis revealed that the mRNA expression levels of *LTBP1* and *TGFBR1* genes were significantly upregulated in the TER group relative to the NOR group. Interestingly, an increase in LTBP1 and TGF-βR1 protein expression was correlated with a decrease in the percentage of normal sperm morphology. Our findings, for the first time, demonstrate a significant association

**Data availability statement:** The LC-MS/MS datasets generated during this study are available in the ProteomeXchange repository under the accession IDs JPST004285 and PXD072557 (via jPOST: https://repository.jpostdb.org/preview/7054576026955c0503b3d8, Access key: 9498). Raw MALDI-TOF MS and clinical data of this study are available from the corresponding author upon reasonable request.

**Funding:** This research was supported by Naresuan University, including the University Income Fund (grant number R2567C018), the Global and Frontier Research University Fund (grant number R2567C003) and the Frontier Research and Innovation Clusters Grant (grant number R2569C004). The funders had no role in study design, data collection and analysis, decision to publish, or preparation of the manuscript.

**Competing interests:** The authors have declared that no competing interests exist.

between the expression of LTBP1 and TGF-βR1 and abnormal sperm morphology, supporting their consideration as novel exploratory biomarkers for further investigation in infertility, especially in TER men.

## Introduction

Until now, most of the infertile population (approximately 50–80 million persons worldwide) has been involved in the male infertility problem [1]. The World Health Organization (WHO) defines the failure to conceive after 12 months of regular, unprotected sexual intercourse as the characteristic of infertility [2]. The cause of most cases of male infertility is unknown, which is classified as idiopathic male infertility [3]. Abnormalities in semen parameters are associated with idiopathic male infertility. Because the major cause of male infertility is unknown, semen analysis is fundamentally used to investigate male infertility. Sperm concentration, motility, and morphology are the main sperm parameters that are used as key indicators in the assessment of male infertility. Men with normal values for these sperm parameters are defined as normozoospermic (NOR) men, whereas men with poor sperm quality are defined by several terms. Teratozoospermic (TER) men are defined as men revealing abnormal sperm morphology, which is one of several terms for men with poor sperm quality. Additionally, poor sperm morphology is a major cause of male infertility, contributing to over 90% of male infertility cases [4]. Infertility clinics frequently encounter men with TER conditions [5]. Considering the unknown causes of male infertility, numerous factors associated with sperm function and fertilisation have been extensively studied. Moreover, due to the lack of information regarding the therapeutic target proteins, there have been concerns about the study of sperm proteins from infertile men.

Proteomic analysis has been used to study the differences in the proteomic profile and identification of seminal plasma and sperm proteins from infertile men [6,7]. More evidence supports the idea that the changes in the protein profile of sperm can adversely affect fertility. Most studies focus on comparing the protein expression profiles of NOR men with those of infertile men exhibiting abnormal sperm parameters, such as asthenozoospermia (characterised by defective sperm motility) and oligoasthenozoospermia (characterised by defects in both sperm count and motility) [8–12]. Notably, a previous study has indicated the differentially expressed proteins (DEPs) in men with TER and oligoasthenoteratozoospermia (characterised by defects in sperm count, motility, and morphology) [13]. Although the sperm protein biomarkers have been widely studied, the entire scope of pathological changes is yet unknown. Several growth factors, such as insulin-like growth factor-1 (IGF1), epidermal growth factor (EGF), and transforming growth factor beta (TGF-β) in human sperm, have been associated with male infertility [14]. The TGF-β signalling pathway consists of many components associated with signal transduction, including TGF-β ligands (TGF-β1, TGF-β2, and TGF-β3), receptors (TGF-β receptor types 1 and 2), and mediators. The interaction of TGF-β ligands with their receptors results in the triggering of a complex intracellular transduction pathway. The TGF-β ligands have been identified in several organs and participate in cell homeostasis and development

[15,16]. TGF-β1 and its receptor (TGF-βR1) are expressed in human Leydig cells, Sertoli cells, spermatogonia, spermatocytes, and sperm cells [17]. The components of the TGF-β signalling pathway play a role in the male reproductive system, especially in seminal plasma and sperm. TGF-β contained in the seminal plasma acts as an immune regulatory cytokine involved in embryonic development and implantation in the female reproductive tract [18]. Even though extensive research has been done on sperm protein biomarkers, the understanding of the proteomic alterations underlying male infertility remains limited, particularly in cases of TER. The potential involvement of the TGF-β signalling pathway in regulating sperm morphology has been hypothesised but not yet clearly elucidated.

Therefore, this study aimed to investigate the proteomic profile of human sperm from TER men, with a focus on identifying key proteins within the TGF-β signalling pathway that may contribute to abnormal sperm morphology.

## Materials and methods

### Semen samples and analysis

Forty-two volunteer patients who obtained semen analysis donated semen samples between December 2017 and September 2018 at the Naresuan Infertility Centre, Faculty of Medicine, Naresuan University, Thailand, for examination. The experimental protocols for this study were reviewed and approved by the Naresuan University Institutional Review Board (IRB No. P1-0012/2565; COE No. 015/2022) and the Human Research Ethics Committee of Walailak University (Approval Nos. WUEC-21-213-01 and WUEC-21-213-02). Written informed consent was provided by each volunteer. The researchers did not have access to identifiable information during or after data collection. Archived semen samples were accessed for proteomic and immunocytochemistry analyses on 15 March 2022 as part of the approved study protocol. The semen preparation and analysis, including sperm morphology, were studied, as previously reported by Kaewman et al. [19]. In this investigation, all semen samples showed normal in macroscopic examination, including appearance, liquefaction, and pH value [20]. They had a sperm concentration ≥ 15 x $10^6$ per ml. The NOR group, which consisted of 20 volunteer patients, had normal values for all parameters, including sperm concentration (≥ 15 x $10^6$ per ml), progressive motility (≥ 32%), and morphology (normal form ≥ 4%). The TER group included 22 volunteer patients who had normal sperm concentration and progressive motility but abnormal sperm morphology (normal form < 4%). For the proteomic, reverse transcription-quantitative polymerase chain reaction, and immunocytochemistry analyses, all semen samples were kept at −80°C.

### Proteomic analysis

Proteomic analysis was conducted using liquid chromatography–tandem mass spectrometry (LC-MS/MS). The human sperm proteins were extracted from each individual semen sample (n = 20 NOR and n = 22 TER). A cold homogenising buffer was used to homogenise sperm samples. After centrifuging, the pellet was homogenised in lysis buffer. The Pierce BCA Protein Assay kit (Thermo Fisher Scientific) was used to quantify protein. Individuals extracted proteins from two groups of samples were examined by proteome analysis. Protein digestion was performed after reducing and alkylating, with incubation for an hour at 56°C and an hour at room temperature, respectively. Ammonium bicarbonate (10 mM) was used to dilute all solutions. In the in-solution digestion, proteins were then treated with 50 ng of trypsin overnight at 37°C. Before resuspending the eluted peptides in 0.1% formic acid, they were dried. The eluted peptides solution was centrifuged before being resuspended in the water used for liquid chromatography–mass spectrometry (LC-MS). Peptide evaluation was carried out using the Impact II UHR-TOF MS system (Bruker Daltonics Ltd.) in conjunction with the nanoLC system (Thermo Fisher Scientific). The protein quantification associated with the Homo sapiens database was carried out using the MaxQuant version 1.6.6.0 software and the Andromeda search engine [21]. Label-free quantitation with MaxQuant's standard settings was applied. For the main search, we allowed for a maximum of two missed cleavages and a mass tolerance of 0.6 daltons. Trypsin and carbamidomethylation of cysteine were used as digestive enzymes and a fixed modification, respectively. Methionine oxidation and the acetylation of the protein N-terminus were used to

achieve variable modifications. Protein identification was restricted to peptides with a minimum of 7 amino acids and at least one unique peptide. We considered only proteins with at least two peptides. The protein-level false discovery rate (FDR) was controlled at 1% based on reversed protein sequences. Five was the maximum number of modifications per peptide. Missing values were handled using Perseus software (version 1.6.6.0) by inputting a constant zero for proteins not detected in all samples, providing a consistent baseline for statistical analysis [22]. Log2 (Intensity) values were used to quantify the expression levels of protein in human sperm.

## Bioinformatics analysis

The total number of proteins identified in NOR and TER groups was analysed using a Venn diagram [23]. The overlapping proteins in human sperm between the NOR and TER groups were also identified. An online tool for gene functional annotation, the Database for Annotation, Visualisation, and Integrated Discovery (DAVID version 6.8; https://david.ncifcrf.gov/), was used to assess pathway enrichment [24]. The Kyoto Encyclopedia of Genes and Genomes (KEGG) pathways were also determined. We employed the UniProt Knowledgebase (UniProt KB; https://www.uniprot.org) to annotate the information of the proteins. Moreover, the interactions between proteins were visualised using the Search Tool for the Retrieval of Interacting Genes (STRING) online tool (https://string-db.org/).

## Reverse transcription-quantitative polymerase chain reaction (RT-qPCR) analysis

Validation analysis was performed via RT-qPCR on sperm samples from 14 NOR and 16 TER men who showed detectable activity in the TGF-β signalling pathway. The FavorPrep™ Tissue Total RNA Mini Kit (Favorgen Biotech Corp, Taiwan) was used to extract total RNA from the sperm samples following the manufacturer's protocol. The mRNA was reverse transcribed into cDNA using the Maxime™ RT PreMix (Oligo dT Primer) (iNtRON Biotechnology, Korea). The primers for the target genes, including the *LTBP1* gene (forward primer 5'-TGAATGCCAGCACCGTCATCTC-3'; reverse primer 5'-CTGGCAAACACTCTTGTCCTCC-3') and the *TGFBR1* gene (forward primer 5'-TCAGCTCTGGTTGGTGTCAG-3'; reverse primer 5'-ATGTGAAGATGGGCAAGACC-3'), were used. The glyceraldehyde-3-phosphate dehydrogenase (*GAPDH*) gene (forward primer 5'-CTCAACGACCACTTTGTCAAGCTCA-3'; reverse primer 5'-GGTCTTACTCCTTG GAGGCCATGTG-3') was used as an internal control to normalise the mRNA expression of the target genes. Each PCR product was amplified from 4 ng of cDNA template using qPCRBIO SyGreen Mix (PCR Biosystems, UK). Quantitative PCR was conducted on a QuantStudio™ 5 Real-Time PCR System (Thermo Fisher Scientific, USA). We used the 2-ΔΔCq method to measure the relative mRNA expression levels of *LTBP1* and *TGFBR1* genes.

## Immunocytochemistry analysis

To determine the TGF-βR1 localization and protein expression, immunocytochemistry was performed on fixed sperm smears. Briefly, sperm were placed on adhesive microscope slides before being air-dried at room temperature. A rabbit polyclonal anti-TGF-βR1 antibody (Abcam; ab31013) was used as a primary antibody for immunoreaction. The immunoreactive signals of TGF-βR1 were amplified by using avidin-biotinylated horseradish peroxidase complexes (Vector). DAB (3,3'-Diaminobenzidine) (Vector) was used to detect TGF-βR1 immunoreactive signals. Before the measurement of the immunoreactive signals of TGF-βR1, the sections were counterstained with hematoxylin and then mounted with histopathology mounting medium. The high-resolution digital images were taken from the immunocytochemistry-stained slides using a microscope camera. The quantitative protein expression of TGF-βR1 was determined from 200 sperm in each sample. ImageJ software (NIH, Bethesda, MD, freely available at https://imagej.nih.gov/ij/) was used for determining the intensity values of immunoreactions in each spermatozoon. The expression levels of TGF-βR1 were quantified as the integrated optical density per area, with values normalised to the control group and represented as relative optical density (ROD).

## Statistical analysis

The normal distribution of the data was estimated using a Shapiro-Wilk test. The statistical difference between the two groups was analysed using a Student's t-test (parametric data). The data were shown as mean±SEM. Statistically significant was considered at p<0.05. Multiple testing correction was performed using the Benjamini-Hochberg procedure, and results were reported as FDR-adjusted p-values (q-values), with a significance threshold set at an FDR of 0.01. Moreover, Pearson's correlation coefficient was used to investigate the correlation between protein expressions in human sperm and the percentage of normal sperm morphology.

## Results

### Proteomic analysis of human sperm from TER and NOR men

A Venn diagram indicated that 1,161 proteins and 1,243 proteins were uniquely expressed in the NOR and TER groups, respectively. A total of 9,088 overlapping proteins were identified between these groups (see Fig 1). The KEGG pathway analysis was used to identify the most relevant biological pathways for the overlapping proteins. The top 5 categories, which were classified based on protein count, were metabolic pathways, herpes simplex virus 1 infection, human papillomavirus infection, PI3K-Akt signalling pathway, and MAPR signalling pathway. This study highlighted the TGF-β signalling pathway, which was identified as one of the top 40 categories within the KEGG pathway analysis. We identified 39 overlapping proteins in human sperm between the TER and NOR groups, associated with the TGF-β signalling pathway (see S1 Table). Their enrichment in the TGF-β signalling pathway is shown in Fig 2.

Six of the 39 overlapping proteins with a log2 fold change ≥ 1 or ≤ −1 were defined as DEPs. Table 1 provides a description of the log2 fold change in these 6 DEPs. E3 ubiquitin-protein ligase SMURF1 (SMURF1) and hemojuvelin (HJV) were downregulated, whereas activin receptor type-2B (ACVR2B), fibrillin-1 (FBN1), mothers against decapentaplegic homolog 6 (SMAD6), and latent-transforming growth factor beta-binding protein 1 (LTBP1) were upregulated. We found that among these proteins, the LTBP1 protein expression was significantly increased in the TER group compared with the NOR group, with a log2 fold change of 1.681 (p<0.05) (see Table 1 and Fig 3). Despite not reaching statistical significance following FDR correction, the LTBP1 exhibited a high fold-change and clear biological relevance in the proteomic screen. To verify this candidate, we performed independent validation using RT-qPCR, which successfully confirmed its differential expression.

Furthermore, the protein–protein interaction network analysis using the STRING database demonstrated the relationships among the 6 DEPs in the TGF-β signalling pathway and their interactions with TGF-βR1 (see Fig 4). Notably, TGF-βR1 and LTBP1 were implicated in TGF-β receptor activity, TGF-β binding, and cytokine binding, as indicated by Gene Ontology (GO) terms related to molecular function.

### Validation of *LTBP1* and *TGFBR1* mRNA expression in human sperm

To validate the results of the proteomic analysis, we quantified the mRNA expression of *LTBP1* and *TGFBR1* genes in the sperm samples. The RT-qPCR results confirmed a significant differential expression of both genes. The relative mRNA expression of these genes was shown in Fig 5. The expression of the *LTBP1* gene in the TER group was significantly increased compared to the NOR group (1.23±0.05 versus 1.03±0.06, p<0.01), which is consistent with the proteomic analysis results. Moreover, the expression of the *TGFBR1* gene in the TER group was also significantly increased compared to the NOR group (1.16±0.02 versus 1.01±0.03, p<0.001).

### TGF-βR1 protein expression in human sperm

Based on the results of immunocytochemistry analysis, the TGF-βR1 protein expression was localised in the human sperm head. The TGF-βR1 immunostaining was detected predominantly at the postacrosomal region and equatorial segment (posterior acrosome). Moreover, the expression was found in some parts of the anterior acrosome (see Fig 6). The

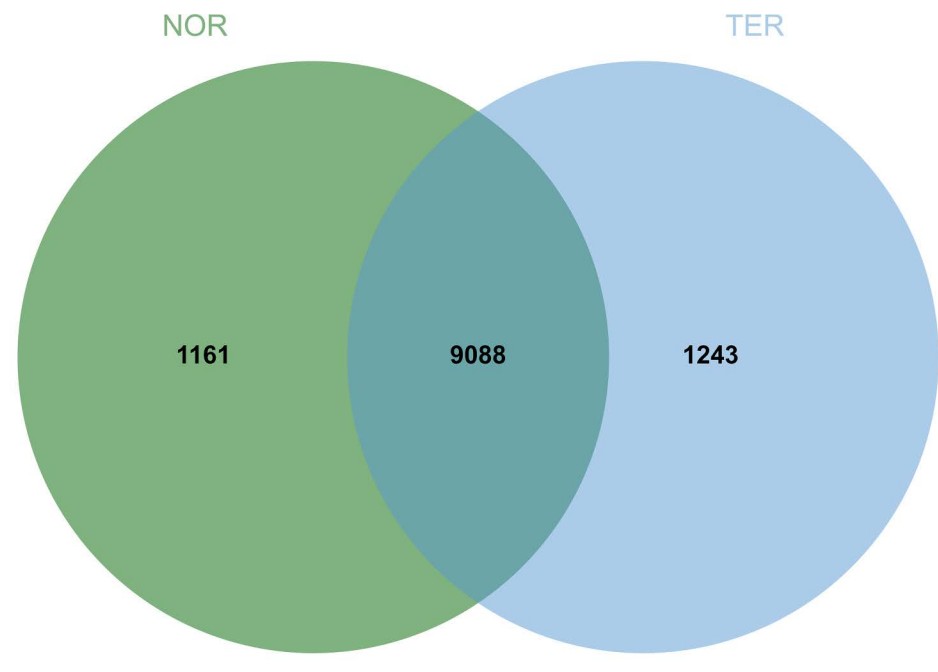

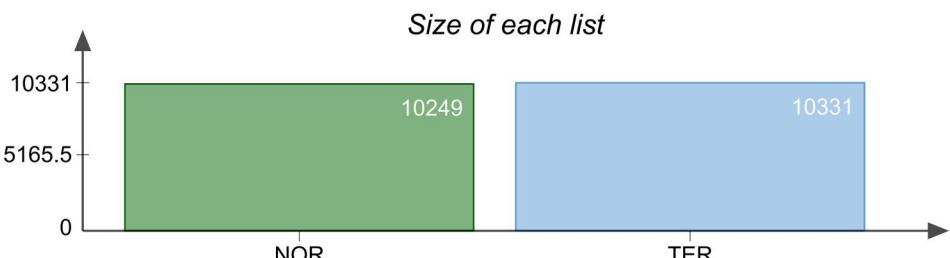

**Fig 1. A Venn diagram showing overlaps between human sperm proteins among the NOR group (green circle) and TER group (blue circle).**

levels of TGF-βR1 protein expression, which were measured in the human sperm head, were significantly increased in the TER group (1.12 ± 0.02) compared with the NOR group (1.00 ± 0.03) (see Fig 7).

### Correlations of LTBP1 and TGF-βR1 protein expression in human sperm with sperm morphology

We analysed the correlation study of the protein expression of LTBP1 and TGF-βR1 with the result of sperm morphology, which has previously been reported in our previous study [19]. The negative correlations between the percentage of normal sperm morphology and the protein expression of LTBP1 (r = −0.4588, p = 0.0108) and TGF-βR1 (r = −0.4851, p = 0.0066) were reported (see Fig 8).

### Discussion

This study extends the proteome data on sperm from infertile men by specifically focusing on TER men. The findings were derived from individual samples, with appropriate sample sizes in each group, including NOR and TER men. According to

**Fig 2. Enrichment of the 39 overlapping proteins (red squares) in the TGF-β signalling pathway in human sperm (TER versus NOR).**

our findings, metabolic pathways were the most relevant of the KEGG pathway discoveries in the overlapping proteins in human sperm between TER and NOR men. In the previous study, three KEGG pathways, including metabolic pathways, endocytosis, and spliceosome, were reported to be among the top 20 of pathway enrichment findings in sperm from infertile men with severe oligoasthenoteratozoospermia [13]. This result being in accordance with our result, we suggest that metabolism may be necessary in maintaining normal sperm function. There was evidence of the relationship between the metabolic syndrome and the TGF-β signalling pathway. Gene-gene interactions within the TGF-β family contribute to the risk of metabolic syndrome, which can impact male infertility [25,26].

**Table 1. The list of the 6 DEPs in the TGF-β signalling pathway in human sperm (TER versus NOR).**

| Accession number | Protein name | Gene symbol | Log 2 fold change[a] (TER/NOR) | p-value | q-value |
|---|---|---|---|---|---|
| Q9HCE7 | E3 ubiquitin-protein ligase SMURF1 | SMURF1 | −2.529 | 0.165 | 0.711 |
| Q6ZVN8 | Hemojuvelin | HJV | −1.075 | 0.059 | 0.570 |
| Q13705 | Activin receptor type-2B | ACVR2B | 1.179 | 0.216 | 0.693 |
| P35555 | Fibrillin-1 | FBN1 | 1.205 | 0.150 | 0.696 |
| O43541 | Mothers against decapentaplegic homolog 6 | SMAD6 | 1.412 | 0.508 | 0.843 |
| **Q14766** | **Latent-transforming growth factor beta-binding protein 1** | **LTBP1** | **1.681** | **0.020** | **0.433** |

[a]Values greater than 0 indicate up-regulation, while values less than 0 indicate down-regulation.

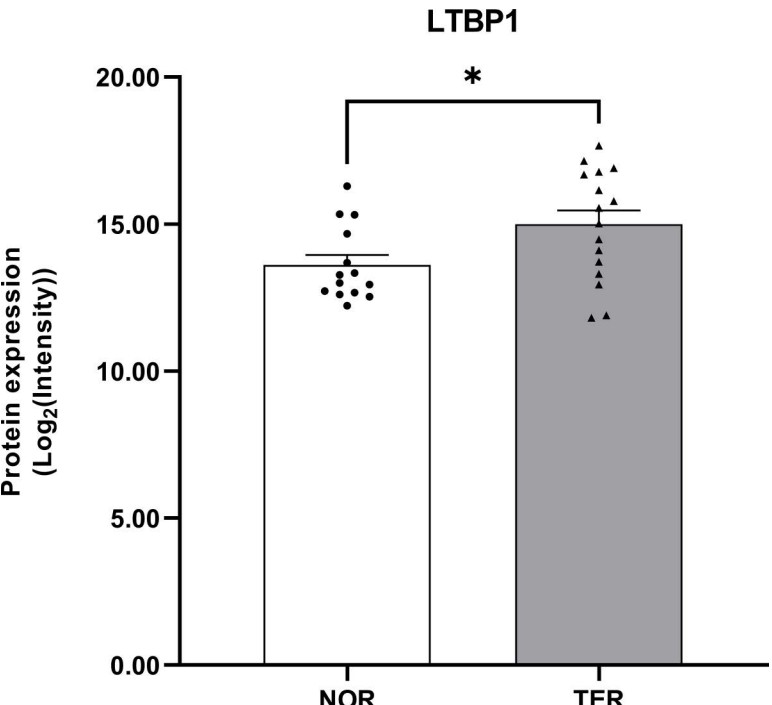

**Fig 3. The LTBP1 protein expression levels in human sperm.** Values are shown as mean ± SEM, TER (n = 16) versus NOR (n = 14). *p < 0.05; unpaired Student's t test.

The quantitative proteomic profiles obtained in this study enabled a comparison of protein expression changes in human sperm. Our results from the KEGG analysis revealed that the overlapping proteins in the sperm from the TER and NOR men were associated with the TGF-β signalling pathway. This pathway involves reproductive functions such as spermatogenesis, sperm maturation, and sperm functions. The members of the TGF-β signalling pathway are subdivided into two functional groups, including the TGF-β and bone morphogenetic protein (BMP) families. The TGF-β family includes TGF-βs, Activins, nodals, and some growth and differentiation Factors (GDFs). TGF-β ligands are cytokines that belong to the peptidic growth factor family. They are present in two forms, including latent or acid-activatable (inactive) and free (active) forms. There are several types of receptors for TGF-β signal transduction. Activin receptor types 1 and 2 mediate signals for activin. Moreover, TGF-βR1 and TGF-βR2 are activated by the ligands TGF-β isoforms 1, 2, and 3,

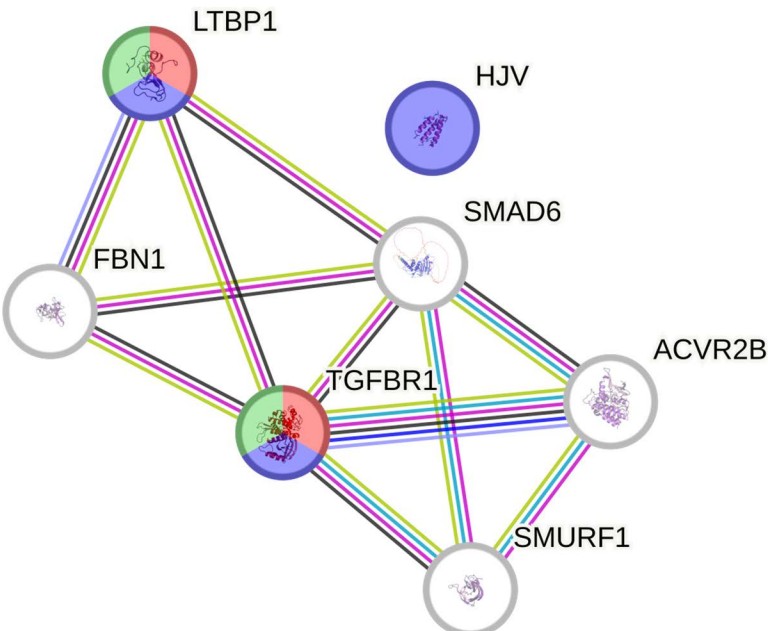

**Fig 4. The STRING protein-protein interaction network.** These proteins are included in the TGF-β signalling pathway. The red, green, and blue colours represent molecular function terms, including TGF-β receptor activity, TGF-β binding, and cytokine binding, respectively.

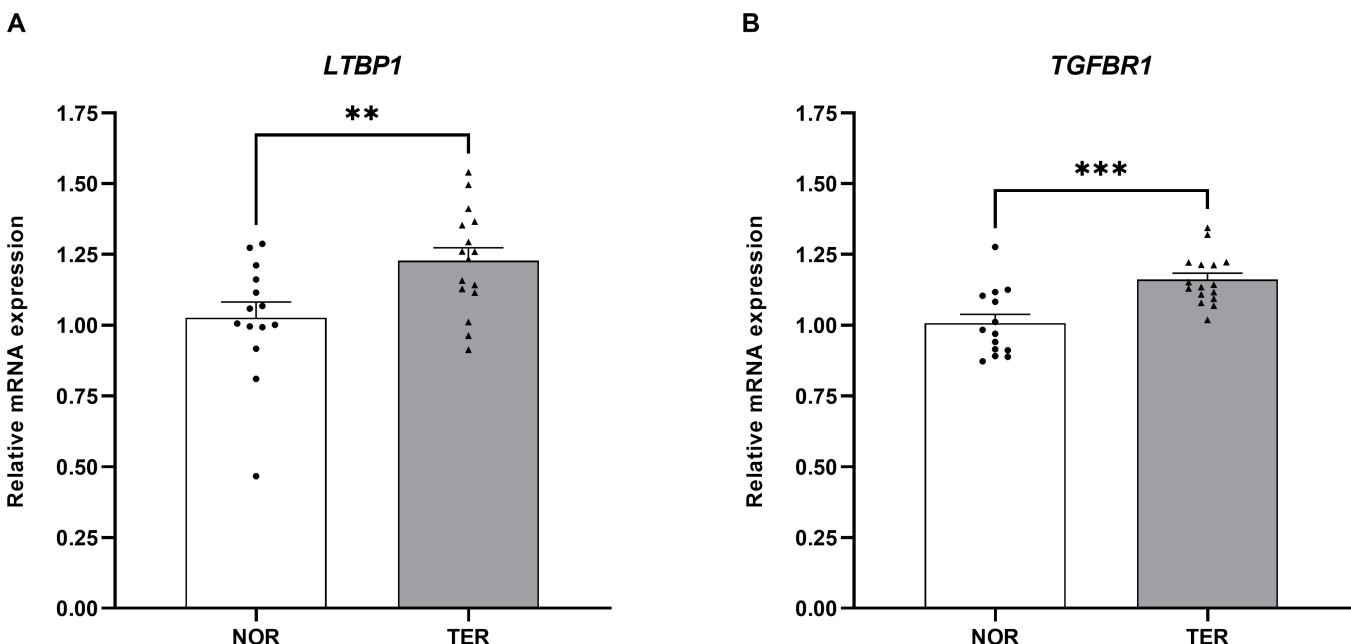

**Fig 5. The mRNA expression of *LTBP1* and *TGFBR1* genes in human sperm.** Values are shown as mean ± SEM, TER (n = 16) versus NOR (n = 14). **p < 0.01 and ***p < 0.001; unpaired Student's t test.

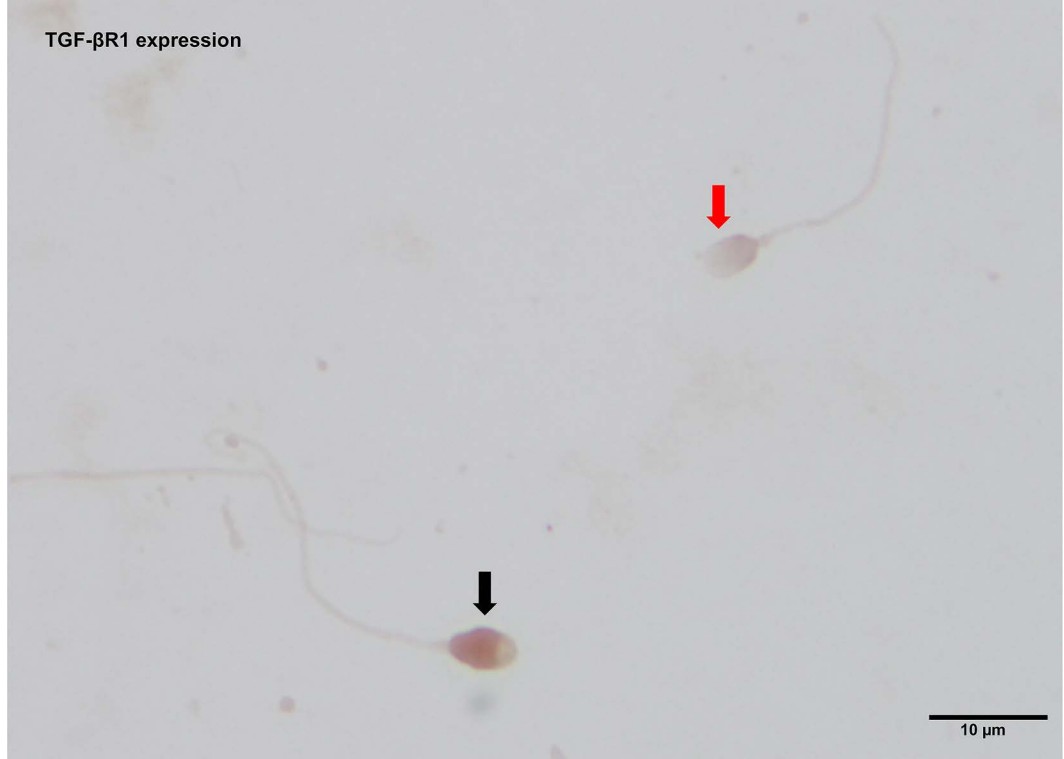

**Fig 6. Immunostaining for TGF-βR1 expression in human sperm at 1000x magnification.** A black arrow indicates an immunopositive spermatozoon, whereas a red arrow indicates an immunonegative spermatozoon.

which trigger signal transduction through TGF-β mediators [27]. The binding proteins, which act as mediators, consist of the latency-associated peptide (LAP) and LTBP1. These binding proteins are important for the activation of active TGF-β release. The TGF-β ligands are secreted into the extracellular medium through the formation of a latent complex. The signal is subsequently transmitted by attaching to their receptors, forming ligand-receptor complexes, and then signalling to the intracellular mediators, known as Smads [28]. Endogenous proteins that bind to and neutralise activins, such as follistatin, may inhibit TGF family ligand signalling. Interestingly, the present study identified 39 overlapping proteins belonging to the TGF-β signalling pathway, as shown in S1 Table. These proteins consist of the TGF-β ligands (TGF-β2 and TGF-β3), activins and their receptors, BMPs and their receptors, regulators (such as follistatin and inhibitory Smads (Smad6 and Smad7)), and mediators (such as ligand-binding proteins).

TGF-β ligands have been identified in several organs. All isoforms of TGF-β have been reported in the male reproductive system, including the testis, epididymis, vas deferens, seminal plasma, and sperm. TGF-β family protein dysfunction has been linked to a variety of human diseases [29,30]. Several studies revealed that the TGF-β signalling pathway is necessary for testicular formation; therefore, it can enhance testicular disease and infertility [31]. It also regulates luteinizing hormone and testosterone production [32]. Moreover, TGF-βs are required for testicular development, normal spermatogenesis, and normal testicular morphology [33]. They control the function of spermatogenic, Sertoli, and Leydig cells through their receptors [34–36]. Under abnormal testicular conditions, the level of TGF-β1 expression in testicular tumors was significantly higher than in normal testicular tissues [37]. Voisin et al. explain the role of TGF-β ligands, particularly TGF-β1 and TGF-β3, in the epididymis, where they have been discovered in different species. The results indicate that the TGF-β ligands act as an inhibitor of epididymal epithelial cell growth to limit uncontrolled proliferation. Their receptors

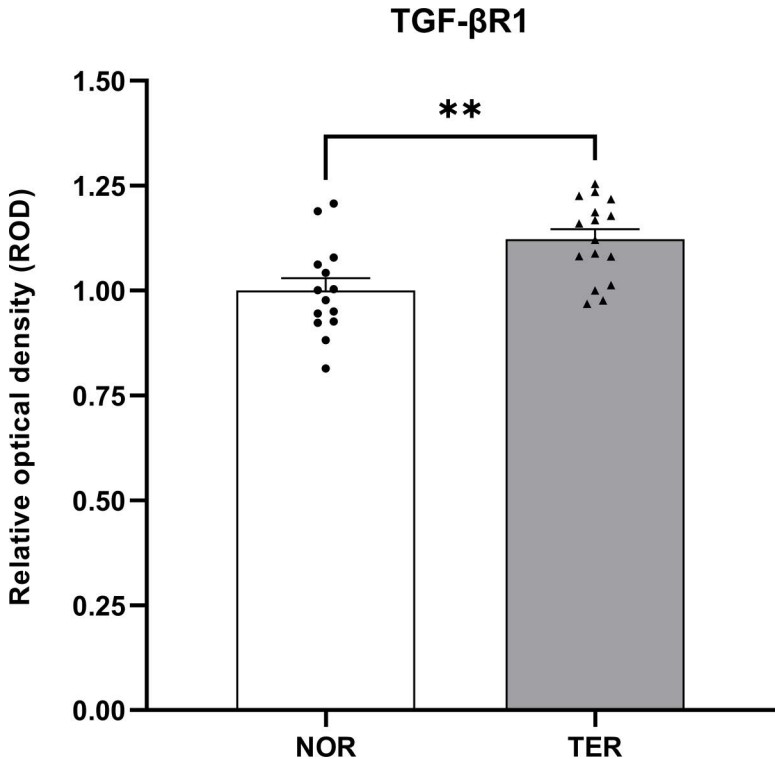

**Fig 7. The TGF-βR1 protein expression in human sperm.** Values are shown as mean ± SEM, TER (n = 16) versus NOR (n = 14). **p < 0.01; unpaired Student's t test.

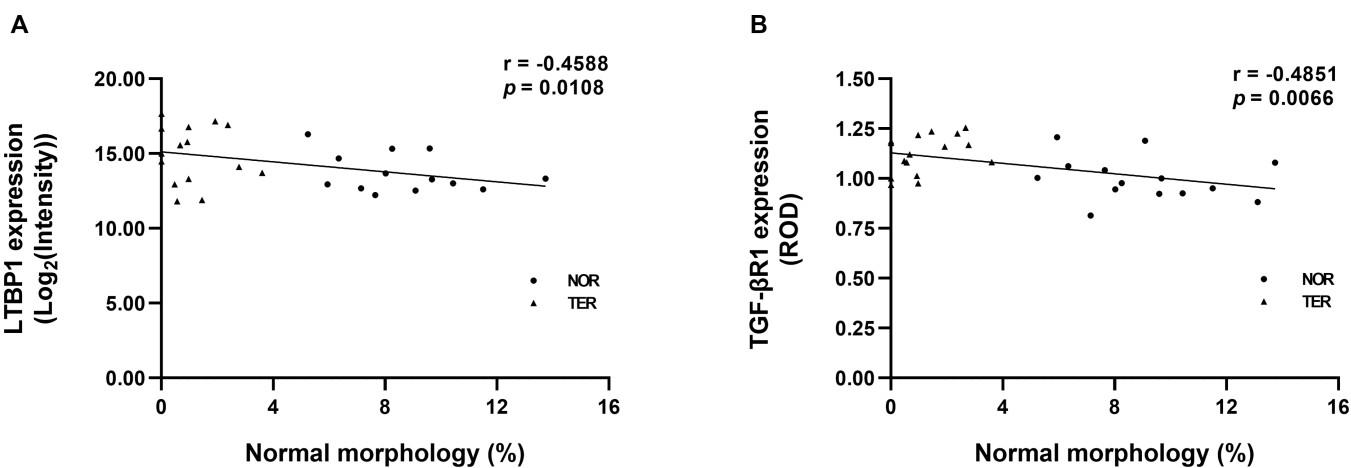

**Fig 8. Correlations between the percentage of normal sperm morphology and the expression of proteins in human sperm. (A)** LTBP1 **(B)** TGF-βR1. A linear regression line (black line) was fitted to all data points.

are found in the principal cells of the same region [38]. Other studies also support the expression of TGF-β isoforms and receptors found in the epididymis and vas deferens. The homeostasis and function of the male reproductive tract are controlled by TGF-β signalling in the epididymis [39,40]. TGF-β1 acts as an immunosuppressive factor in the human seminal

plasma [41]. However, the TGF-β1 concentration in seminal plasma was not significantly different between NOR and infertile men [42]. TGF-β1 has been reported to localise in the posterior acrosome, the neck, and the middle of the tail [43]. In the present study, although changes in TGF-βR1 protein expression in human sperm were not detected by the proteomic analysis, they were observed using immunocytochemistry. We revealed that its localisation was detected predominantly in the human sperm head, especially in the postacrosomal region and equatorial segment (posterior acrosome). Our results, together with previous findings, indicate that the TGF-β ligands and their receptors were also localised in the same region, which is in the sperm head. However, the localisation of TGF-β ligands was not detected at the acrosomal region of the head, which is consistent with the localisation of the TGF-βR1 that was found in the present study. It might be suggested in parallel with other studies that TGF-β ligands may act as an immunosuppressive factor to protect the integrity and morphology of human sperm, but they may not involve acrosome reaction. Moreover, a previous study reported that infertile men with reduced levels of TGF-β often exhibit increased sperm DNA fragmentation [44]. TER men exhibited higher levels of sperm DNA fragmentation and inflammatory markers compared to other infertile men [45]. In this study, the significant upregulation of TGF-βR1 at both the mRNA and protein levels in TER men points toward TGF-β-mediated immune activation, potentially reflecting a response to the higher prevalence of abnormal sperm morphology. This hypothesis can be supported by our finding of a negative correlation between its expression and the percentage of normal sperm morphology. In addition, the expression of TGF-βR1 also enhances sperm quality and serum sex hormone levels. The TGF-βR1 blocker can significantly increase sperm count and motility [17]. Because TGF-β ligands act as immunosuppressive factors, the activity of TGF-β through its receptors (such as TGF-βR1) was significantly higher in the sperm to compensate for the immune activation, resulting in the maintenance of sperm functions.

Other components of the TGF-β family have been reported in several studies. The latent TGF-β binding proteins (LTBPs) are large multidomain glycoproteins that are important for the activation of TGF-β secretion and extracellular TGF-β function. LTBP1 is the only isoform in the LTBP family that can bind to all TGF-β isoforms, including TGF-β1, TGF-β2, and TGF-β3. The complex of TGF-β and LAP is referred to as a small latent complex. Its interactions with TGF-β ligands enhance the complex's folding and secretion into the extracellular matrix. LTBPs also assist in the conversion of latent TGF-β to free active TGF-β. The accumulation and inappropriate activation of latent TGF-β are controlled by LTBP-mediated matrix incorporation [46]. According to previous studies, LTBP participates in TGF-β production, release, and activation [46,47]. The levels of TGF-β and its ability to target correctly in the extracellular matrix are important in the signal regulation of TGF-β functions [48]. The changes in LTBP levels result in defects in several organ systems, such as cardiovascular and lung defects [46]. The transgenic study indicated that LTBP1 overexpression was related to the increase in TGF-β activity [49]. Moreover, the study in the absence of LTBP expression strongly supports a role for LTBP in TGF-β bioavailability and activation [50,51]. Increased latent and active TGF-β levels occurred along with overexpression of LTBP1. These results support the important role of LTBP1 in controlling TGF-β activity. We also observed a significant upregulation in LTBP1 expression at both mRNA and protein levels in the sperm of men with TER, which inversely correlated with the percentage of normal sperm morphology. While the association between upregulation of TGF-βR1 and LTBP1 and abnormal sperm morphology is compelling, it is important to explicitly acknowledge that our study did not account for certain unmeasured confounders, such as subclinical localised inflammation and individual lifestyle factors. These factors could potentially influence TGF-β signalling and sperm quality. However, the study of LTBP1 and TGF-βR1 as biomarkers in larger populations and mechanistic studies exploring their roles affecting sperm morphology will strengthen the translational potential of these findings. We conclude that sperm proteins, such as TGF-family proteins that are specifically relevant to reproductive function, play an important role in sperm function. The effect of TGF-β, one of the anti-inflammatory cytokines, in sperm might increase to regulate the inflammatory response during the fertilisation process, which occurs highly in infertile men. Currently, the proteomic profile of human sperm is very interesting to study for the evaluation of disease progression in various human diseases. Male infertility is one of the important clinical areas of interest. Understanding the proteomic profile of sperm cells in male infertility may be essential to distinguishing individuals

based on their infertility conditions and for identifying biomarkers useful in male fertility diagnosis [52]. Current knowledge of proteomic profiles in human sperm is still limited, especially regarding sperm samples derived from infertile men. Alterations in the proteomic profile have been reported in asthenozoospermia and oligozoospermia but not in TER [10].

## Conclusion

Overall, our findings compare the protein profiles of human sperm from TER and NOR men. The TGF-β signalling pathway was the focus of the KEGG pathways identified through analysis of overlapping proteins between the two groups. Thirty-nine overlapping proteins in human sperm were identified as members of the TGF-β signalling pathway. Among them, LTBP1 was the most predominantly altered protein in the TER men. Validation of *LTBP1* and *TGFBR1* genes, key components of the TGF-β signalling pathway, revealed a significant increase in their mRNA expression levels in the sperm of TER men. Furthermore, the results from immunocytochemistry analysis indicated a significant increase of TGF-βR1 protein in human sperm from TER men compared to NOR men. An increase in LTBP1 and TGF-βR1 expression in human sperm is associated with a decrease in the percentage of normal sperm morphology. This result indicates that these proteins may be involved in the mechanism of spermatogenesis. We suggest that LTBP1 and TGF-βR1 proteins have the most impact as sperm protein biomarkers defining infertility related to abnormal sperm morphology. Notably, our findings provide novel evidence supporting the investigation of proteomic profiles within the TGF-β signalling pathway, highlighting their potential relevance as sperm biomarkers for infertility, particularly in the case of TER. They are primarily used to estimate an individual's fertility potential. The insights in the present study will further help in investigating the pathogenic mechanisms of male infertility.

## Supporting information

**S1 Table. The list of the 39 overlapping proteins in the TGF-β signalling pathway in human sperm (TER versus NOR).**
(DOCX)

## Acknowledgments

We appreciate the facilities support from Naresuan University, Walailak University, and the National Center for Genetic Engineering and Biotechnology, National Science and Technology Development Agency, Thailand. The authors are grateful to the staff of the Naresuan Infertility Centre for providing help with sample collection.

## Author contributions

**Conceptualization:** Sutisa Nudmamud-Thanoi, Samur Thanoi.

**Data curation:** Paweena Kaewman, Patcharada Amatyakul, Sittiruk Roytrakul.

**Formal analysis:** Paweena Kaewman, Sittiruk Roytrakul.

**Funding acquisition:** Paweena Kaewman, Sutisa Nudmamud-Thanoi, Samur Thanoi.

**Investigation:** Paweena Kaewman, Sutisa Nudmamud-Thanoi, Sittiruk Roytrakul.

**Methodology:** Paweena Kaewman, Chuchard Punsawad, Sittiruk Roytrakul.

**Project administration:** Paweena Kaewman, Sutisa Nudmamud-Thanoi, Samur Thanoi.

**Resources:** Patcharada Amatyakul, Chuchard Punsawad, Sittiruk Roytrakul.

**Supervision:** Sutisa Nudmamud-Thanoi, Samur Thanoi.

**Validation:** Paweena Kaewman, Samur Thanoi.

**Writing – original draft:** Paweena Kaewman.

**Writing – review & editing:** Sutisa Nudmamud-Thanoi, Samur Thanoi.

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
