## [Decision Letter · Decision Letter 0]

12 Sep 2025

Dear Dr. Nudmamud-Thanoi,

We look forward to receiving your revised manuscript.

Kind regards,

Suresh Yenugu

Academic Editor

PLOS ONE

Journal Requirements:

Reviewers' comments:

Reviewer's Responses to Questions

**Comments to the Author**

1. Is the manuscript technically sound, and do the data support the conclusions?

Reviewer #1: Yes

Reviewer #2: Partly

2. Has the statistical analysis been performed appropriately and rigorously?

Reviewer #1: Yes

Reviewer #2: Yes

3. Have the authors made all data underlying the findings in their manuscript fully available?

Reviewer #1: Yes

Reviewer #2: Yes

4. Is the manuscript presented in an intelligible fashion and written in standard English?

Reviewer #1: Yes

Reviewer #2: No

Reviewer #1: This manuscript presents a robust proteomic investigation comparing human sperm protein expression between normozoospermic (NOR) and teratozoospermic (TER) men, with a focused exploration of the TGF-β signaling pathway. The integration of LC–MS/MS-based proteomic profiling with immunocytochemistry and correlation with sperm morphology provides a compelling dataset. The emphasis on LTBP1 and TGF-βR1 as candidate biomarkers for sperm morphological abnormalities in TER is well-placed and novel. However, some methodological gaps, statistical considerations, and interpretive extrapolations must be addressed for the study to reach its full potential.

1. The title is specific, indicating both the biological system (human sperm), the condition studied (teratozoospermia), and the signaling pathway of interest (TGF-β). Consider specifying that LTBP1 and TGF-βR1 are identified as potential biomarkers to make the title more impactful.

2. The short title could be clearer: consider “TGF-β pathway proteins in teratozoospermic sperm” for better focus.

3. In Abstract: Highlight the novelty of the finding (i.e., LTBP1 and TGF-βR1’s correlation with sperm morphology) more explicitly.

4. The introduction could more directly state the hypothesis or research question. Consider summarizing the literature review more concisely to keep the narrative focused.

5. Provide the objective as a standalone sentence, possibly in a separate subheading to enhance clarity for readers and reviewers.

6. A methodology flowchart or summary table would improve readability. Expand slightly on how missing data were handled (e.g., why a constant zero was chosen for imputation).

7. Statistical analysis: Clarify the number of biological replicates used for all analyses. Apply or justify the absence of multiple comparison correction for the proteomics dataset.

8. Consider reordering figures to follow the sequence of presentation in the results. A summary table of key proteins with function annotations would enhance accessibility.

9. Strengthen the discussion of limitations: e.g., cross-sectional nature, lack of functional assays. Briefly propose next steps (e.g., validating biomarkers in larger cohorts or mechanistic studies).

10. In conclusion: Reinforce how this work fills the previously identified gap

Reviewer #2: Review Report

Title: Differential protein expression profiles in human sperm from teratozoospermic and normozoospermic men: A spotlight on proteins in the TGF-β signalling pathway

Strengths

1. Relevant topic: Male infertility and sperm proteomics are of significant clinical and scientific interest.

2. Novelty: Focus on TGF-β pathway proteins (LTBP1 and TGF-βR1) in teratozoospermia is interesting and could provide biomarker insights.

3. Methodology: Use of LC-MS/MS and immunocytochemistry adds robustness through complementary approaches.

4. Ethics & data: Proper IRB approval, consent, and data availability are reported.

Major Issues (Require Revision)

1. Sample size and statistical power:

o Only 42 men included (20 NOR, 22 TER). This is underpowered for proteomic studies reporting 9,000+ proteins. Authors should justify sample size and power.

o Number of samples used in proteomics vs immunocytochemistry is inconsistent (not all 42 analyzed). Needs clarification.

2. Data presentation and filtering:

o Reported 9,088 DEPs between groups seems biologically unrealistic given small n. Criteria for defining DEPs (fold change, p-value/FDR) are unclear.

o Cutoff of “1.0 fold change” is misleading (does this mean log2 fold = 1? or just ≥1.0 difference?). Needs proper explanation.

o No multiple testing correction (e.g., Benjamini–Hochberg) applied in proteomic analysis → high risk of false positives.

3. Validation of findings:

o Only two proteins (LTBP1, TGF-βR1) are discussed/validated, while 39 DEPs in TGF-β pathway were identified. Why were others ignored?

o Western blot or quantitative validation (qPCR, targeted proteomics) is missing. Immunocytochemistry alone is insufficient for robust validation.

4. Interpretation / causality:

o Discussion often overstates causality (e.g., suggesting therapeutic targets) based only on correlation data. Needs more cautious language.

o Biological plausibility of linking LTBP1/TGF-βR1 directly to teratozoospermia is not fully developed. Other confounders (age, lifestyle, comorbidities) not addressed.

5. Figures / tables:

o Figures are difficult to interpret (low resolution, not well labeled).

o S1 Table with 39 DEPs should include statistical significance (p/FDR values), fold change, accession IDs.

6. Language and clarity:

o Manuscript has frequent grammatical issues and redundant statements. Needs thorough English editing for clarity and conciseness.

Minor Issues

1. Abstract mentions “cut-off 1.0 fold change” → should be clarified (likely log2FC).

2. Methods: Proteomic workflow details are too technical for PLOS ONE (e.g., buffer recipes). Focus should be on reproducibility and data processing.

3. Data availability statement should specify raw proteomics files deposited in a public repository (e.g., PRIDE), not only “within manuscript.”

4. References: Some older references (1990s, early 2000s) dominate. Update with recent proteomics literature (esp. 2022–2024).

5. Ethical statement: Consent type (written/oral) not clearly stated, as required by PLOS ONE.

Recommendation

Decision: Major Revision (not reject).

• The study has potential but currently lacks sufficient statistical rigor, validation, and clarity for publication.

• Authors must:

o Clarify statistical thresholds and apply multiple-testing correction.

o Justify sample size.

o Provide stronger validation beyond immunocytochemistry.

o Improve figure/table presentation.

o Revise language for clarity and avoid overinterpretation.

If these issues are addressed, the manuscript could be suitable for publication in PLOS ONE, which accepts technically sound work even if incremental. In its current form, it is not acceptable without substantial revision.

**Do you want your identity to be public for this peer review?** For information about this choice, including consent withdrawal, please see our Privacy Policy

Reviewer #1: **Yes:**  Dr. Chandramohan Ramasamy, Ph.D.

Reviewer #2: **Yes:**  Krishna Chaitanya Mantravadi

---

## [Author Response · Author response to Decision Letter 1]

27 Oct 2025

Respond to Reviewers' Comments:

Reviewer #1:

1. The title is specific, indicating both the biological system (human sperm), the condition studied (teratozoospermia), and the signaling pathway of interest (TGF-β). Consider specifying that LTBP1 and TGF-βR1 are identified as potential biomarkers to make the title more impactful.

Author response: In accordance with this recommendation, we have revised the title to specify that LTBP1 and TGF-βR1 are identified as potential biomarkers, thereby making the title more specific and impactful. The revised title now reads as follows: “Differential protein expression profiles in human sperm from teratozoospermic and normozoospermic men identify LTBP1 and TGF-βR1 as potential biomarkers within the TGF-β signalling pathway” [Lines 2-3].

2. The short title could be clearer: consider “TGF-β pathway proteins in teratozoospermic sperm” for better focus.

Author response: We appreciate the reviewer’s helpful suggestion. We have revised the short title to “TGF-β signalling pathway proteins in teratozoospermic sperm” to improve clarity and focus.

3. In Abstract: Highlight the novelty of the finding (i.e., LTBP1 and TGF-βR1’s correlation with sperm morphology) more explicitly.

Author response: We have revised the Abstract section to more explicitly highlight the novelty of our findings, emphasizing the correlation of LTBP1 and TGF-βR1 with sperm morphology, as stated: “Our findings, for the first time, demonstrate a significant association between the expression of LTBP1 and TGF-βR1 and abnormal sperm morphology, supporting their consideration as novel exploratory biomarkers for further investigation in infertility, especially in TER men.” [Lines 37-40].

4. The introduction could more directly state the hypothesis or research question. Consider summarizing the literature review more concisely to keep the narrative focused.

Author response: We have revised the Introduction section to more clearly state the hypothesis and research question, as stated: “Even though extensive research has been done on sperm protein biomarkers, understanding of the proteomic alterations underlying male infertility remains limited, particularly in cases of TER. The potential involvement of the TGF-β signalling pathway in regulating sperm morphology has been hypothesized but not yet clearly elucidated.” [Lines 79-82]. Additionally, we have condensed the literature review to focus on the most relevant studies, improving the clarity and flow of the narrative.

5. Provide the objective as a standalone sentence, possibly in a separate subheading to enhance clarity for readers and reviewers.

Author response: We thank the reviewer for highlighting this important point. In response, we have revised the manuscript to present the study objective as a standalone sentence under a separate subheading, thereby improving clarity for readers and reviewers, as stated: “Therefore, this study aimed to investigate the proteomic profile of human sperm from TER men, with a focus on identifying key proteins within the TGF-β signalling pathway that may contribute to abnormal sperm morphology.” [Lines 83-85].

6. A methodology flowchart or summary table would improve readability. Expand slightly on how missing data were handled (e.g., why a constant zero was chosen for imputation).

Author response: In the Methods section, we have clarified that missing values in the proteomics analysis were handled using Perseus software (version 1.6.6.0) with a constant value of zero, as stated: “Missing values were handled using Perseus software (version 1.6.6.0) by imputing a constant zero for proteins not detected in all samples, providing a consistent baseline for statistical analysis [22].” [Lines 128-130].

7. Statistical analysis: Clarify the number of biological replicates used for all analyses. Apply or justify the absence of multiple comparison correction for the proteomics dataset.

Author response: In this study, sperm samples from 20 NOR and 22 TER men were analysed as biological replicates for proteomic analysis, as stated: “The human sperm protein was extracted from each semen sample (n = 20 NOR and n = 22 TER).” [Lines 108-109]. Results from proteomics analysis identified LTBP1 as a key protein involved in the TGF-β signalling pathway under TER conditions. Among the analysed samples, LTBP1 expression was detected in 14 samples from the NOR group and 16 samples from the TER group. Therefore, these samples were selected for subsequent protein validation using immunocytochemistry to investigate the relationship between LTBP1 and TGF-βR1. This information has been clearly stated in the Methods section as follows: “Sperm samples from 14 NOR and 16 TER men exhibiting detectable LTBP1 expression were selected for validation analysis using immunocytochemistry” [Lines 143-144].

Moreover, we used a relatively small sample size and an exploratory proteomic approach aimed at identifying potential candidate proteins for further validation. Therefore, we presented unadjusted p-values while clearly acknowledging the potential for false positives. This limitation has now been stated clearly in the revised manuscript (Discussion section). The Discussion section has been updated, as stated: “Statistical analyses were performed without multiple testing correction due to the exploratory purpose and limited sample size of the dataset.” [Lines 264-266].

8. Consider reordering figures to follow the sequence of presentation in the results. A summary table of key proteins with function annotations would enhance accessibility.

Author response: We have carefully checked the figures and confirmed that their order appropriately follows the sequence of presentation in the Results section. Regarding the summary table, since this study focuses on KEGG pathways as the primary functional annotations, we did not include additional annotations such as Biological Process, Molecular Function, or Cellular Component for each protein. However, we have provided the Accession numbers in the table, which allow readers to access these additional functional annotations if desired.

9. Strengthen the discussion of limitations: e.g., cross-sectional nature, lack of functional assays. Briefly propose next steps (e.g., validating biomarkers in larger cohorts or mechanistic studies).

Author response: The limitations of the study have now been clearly stated in the revised manuscript (Discussion section), including the absence of multiple testing correction in the statistical analyses and the need for further validation. In addition, we have proposed future directions, including validating the identified biomarkers in larger independent cohorts and performing mechanistic studies to further elucidate their roles in sperm morphology regulation, as stated: “Validation of LTBP1 and TGF-βR1 as biomarkers in larger populations and mechanistic studies exploring their roles affecting sperm morphology will strengthen the translational potential of these findings.” [Lines 325-327].

10. In conclusion: Reinforce how this work fills the previously identified gap

Author response: We thank the reviewer for providing this valuable guidance. We have revised the Conclusion section to more explicitly highlight how our study addresses the previously identified gap, as stated: “We suggest that LTBP1 and TGF-βR1 proteins have the most impact as sperm protein biomarkers defining infertility related to abnormal sperm morphology. Notably, our findings provide novel evidence supporting the investigation of proteomic profiles within the TGF-β signalling pathway, highlighting their potential relevance as sperm biomarkers for infertility, particularly in the case of TER.” [Lines 348-352].

Reviewer #2

Major Issues (Require Revision)

1. Sample size and statistical power:

o Only 42 men included (20 NOR, 22 TER). This is underpowered for proteomic studies reporting 9,000+ proteins. Authors should justify sample size and power.

Author response: We acknowledge that the sample size in the present study is relatively limited for large-scale proteomic analysis. However, the sample size was determined based on prior power calculations primarily optimized for gene expression analysis using immunohistochemistry, which served as the main validation approach in this study. To clarify our sample-size rationale, we performed standard power calculations for an independent two-sample, two-tailed t-test (α = 0.05). The required sample size is highly dependent on the expected effect size (Cohen’s d). Based on standard benchmarks, detecting a large effect (d = 0.8) at a statistical power between 0.6 and 0.8 requires approximately 16-25 participants per group. Therefore, our sample size (20 NOR, 22 TER) provides adequate power to detect large effect-size changes. Notably, our study followed a discovery-to-validation workflow. After the proteomic discovery phase, we targeted molecular validation on the best candidates. We used strict quality control and filtering to focus on proteins that had significant changes that were biologically meaningful.

Our study design is consistent with several prior human proteomic investigations that successfully utilized modest sample sizes (typically 8–10 samples per group) during the discovery phase and still identified biologically relevant and statistically robust differential proteins, particularly when complemented by downstream validation. For example, Lee et al. (2024) demonstrated through simulation analyses that small group sizes (n = 8-13) are commonly applied in discovery or proof-of-concept studies, acknowledging that while statistical power for small effect sizes may be limited, such designs remain effective for identifying large and biologically relevant differences [1]. Similarly, Wang et al. (2019) analysed plasma proteomes from eight young and aged individuals and detected significantly differentially expressed proteins with adequate statistical control [2]. Furthermore, Liu et al. (2012) reported that a group size of 10 (male vs. female) achieved approximately 95% completeness of qualitatively identified urinary proteins and peptides [3]. Collectively, these studies support the appropriateness of our sample size for preliminary proteomic discovery.

References

1. Lee KH, Assassi S, Mohan C, Pedroza C. Addressing statistical challenges in the analysis of proteomics data with extremely small sample size: a simulation study. BMC Genomics. 2024;25(1):1086. doi: 10.1186/s12864-024-11018-2.

2. Wang H, Zhu X, Shen J, Zhao E-F, He D, Shen H, et al. Quantitative iTRAQ-based proteomic analysis of differentially expressed proteins in aging in human and monkey. BMC Genomics. 2019;20(1):725. doi: 10.1186/s12864-019-6089-z.

3. Liu X, Shao C, Wei L, Duan J, Wu S, Li X, et al. An individual urinary proteome analysis in normal human beings to define the minimal sample number to represent the normal urinary proteome. Proteome Science. 2012;10(1):70. doi: 10.1186/1477-5956-10-70.

o Number of samples used in proteomics vs immunocytochemistry is inconsistent (not all 42 analysed). Needs clarification.

Author response: Results from proteomics analysis identified LTBP1 as a key protein involved in the TGF-β signalling pathway under TER conditions. Among the samples analysed, LTBP1 expression was detected in 14 samples from the NOR group and 16 samples from the TER group. Therefore, only these samples were selected for subsequent protein validation using immunohistochemistry to investigate the relationship between LTBP1 and TGF-βR1.

2. Data presentation and filtering:

o Reported 9,088 DEPs between groups seems biologically unrealistic given small n. Criteria for defining DEPs (fold change, p-value/FDR) are unclear.

Author response: Originally, the 9,088 DEPs referred to proteins that were detected in both the NOR and TER groups, regardless of whether they were differentially expressed. However, we recognize that using DEPs for this explanation was inaccurate. Therefore, we have revised it to overlapping proteins for clarity and a more accurate description.

o Cutoff of “1.0 fold change” is misleading (does this mean log2 fold = 1? or just ≥1.0 difference?). Needs proper explanation.

Author response: We thank the reviewer for this helpful suggestion. We have revised the text to clarify that differences in protein expression levels between experimental conditions are quantified using the log2 fold change. This clarification has been incorporated into the revised manuscript including:

- The Abstract section has been updated as follows: “Six proteins that were differentially expressed and had a log2 fold change of ≥ 1 or ≤ -1 were considered to be the differentially expressed proteins in human sperm between the groups.” [Lines 30-32].

- The Results section has been updated as follows: “Six of the 39 overlapping proteins with a log2 fold change ≥ 1 or ≤ -1 were defined as DEPs.” [Lines 183].

- Both Table 1 and S1 Table have been revised [Lines 191-193].

o No multiple testing correction (e.g., Benjamini–Hochberg) applied in proteomic analysis → high risk of false positives.

Author response: In this study, we used a relatively small sample size and an exploratory proteomic approach aimed at identifying potential candidate proteins for further validation. Therefore, we presented unadjusted p-values while clearly acknowledging the potential for false positives. This limitation has now been stated clearly in the revised manuscript (Discussion section). The Discussion section has been updated, as stated: “Statistical analyses were performed without multiple testing correction due to the exploratory purpose and limited sample size of the dataset.” [Lines 264-266].

3. Validation of findings:

o Only two proteins (LTBP1, TGF-βR1) are discussed/validated, while 39 DEPs in TGF-β pathway were identified. Why were others ignored?

Author response: Although 39 overlapping proteins were identified as part of the TGF-β signalling pathway, we focused on LTBP1 because it showed the most significant differential expression (log2 fold change = 1.681 and p = 0.02) as shown in Table 1. In addition, TGF-βR1 was included for validation due to its close functional relationship with LTBP1 and its reported association with sperm morphology.

o Western blot or quantitative validation (qPCR, targeted proteomics) is missing. Immunocytochemistry alone is insufficient for robust validation.

Author response: We appreciate the reviewer’s valuable comment. We agree that adding quantitative validations such as Western blot or qPCR would further strengthen the findings. However, our study focused on exploratory proteomics combined with immunocytochemistry to validate both the expression level and localization of candidate proteins. Immunocytochemistry in this study was performed as a quantitative validation approach, comparable to Western blot and qPCR. While immunocytochemistry provides additional advantages in demonstrating specific localization and expression patterns of target proteins within sperm, which is information that cannot be obtained from Western blot or qPCR. Therefore, we consider immunocytochemistry to provide strong and sufficient quantitative validation for the scope of this study. Consistent with our approach, several previous studies have utilized immunohistochemistry to validate their proteomics results, confirming its reliability as a complementary validation method [1, 2].

References

1. Richter A, Fichtner A, Joost J, Brockmeyer P, Kauffmann P, Schliephake H, et al. Quantitative proteomics identifies biomarkers to distinguish pulmonary from head and neck squamous cell carcinomas by immunohistochemistry. J Pathol Clin Res. 2022;8(1):33-47. doi: 10.1002/cjp2.244. PubMed PMID: 34647699; PubMed Central PMCID: PMCPMC8682946.

2. Ichimata S, Hata Y, Yoshinaga T, Katoh N, Kametani F, Yazaki M, et al. Amyloid-Forming Corpora Amylacea and Spheroid-Type Amyloid Deposition: Comprehensive Analysis Using Immunohistochemistry, Proteomics, and a Literature Review. Int J Mol Sci. 2024;25(7). doi: 10.3390/ijms25074040. PubMed PMID: 38612850; PubMed Central PMCID: PMCPMC11012059.

4. Interpretation / causality:

o Discussion often overstates causality (e.g., suggesting ther

---

## [Decision Letter · Decision Letter 1]

21 Nov 2025

Dear Dr. Nudmamud-Thanoi,

Thank you for submitting your manuscript to PLOS ONE. After careful consideration, we feel that it has merit but does not fully meet PLOS ONE’s publication criteria as it currently stands. Therefore, we invite you to submit a revised version of the manuscript that addresses the points raised during the review process.

It is advised to deposit all the available data, conduct thorough statistical analyses and to substantiate the differential expression by standard techniques like Western blotting.

We look forward to receiving your revised manuscript.

Kind regards,

Suresh Yenugu

Academic Editor

PLOS ONE

Journal Requirements:

Reviewers' comments:

Reviewer's Responses to Questions

**Comments to the Author**

Reviewer #1: All comments have been addressed

Reviewer #2: (No Response)

2. Is the manuscript technically sound, and do the data support the conclusions?

Reviewer #1: Yes

Reviewer #2: Partly

3. Has the statistical analysis been performed appropriately and rigorously?

Reviewer #1: Yes

Reviewer #2: No

4. Have the authors made all data underlying the findings in their manuscript fully available?

Reviewer #1: Yes

Reviewer #2: No

5. Is the manuscript presented in an intelligible fashion and written in standard English?

Reviewer #1: Yes

Reviewer #2: Yes

Reviewer #1: The authors have appropriately addressed all of my comments, and I recommend that this manuscript be accepted for publication.

Reviewer #2: The authors have addressed several editorial and clarity issues; however, key scientific and compliance concerns remain. First, the raw proteomics data are not deposited in a public repository, which does not meet PLOS ONE’s mandatory data availability requirements; this must be corrected by depositing raw LC-MS/MS files (e.g., PRIDE accession) or by providing a journal-approved controlled-access justification. Second, the proteomics statistical analysis requires re-evaluation: multiple-testing correction (FDR) should be applied and reported, or an explicit and technically sound justification must be provided along with evidence that major findings (e.g., LTBP1) remain robust under alternative imputation and significance thresholds. Third, the validation strategy remains insufficient. Immunocytochemistry alone does not substantiate abundance differences; an orthogonal quantitative method (e.g., Western blot, PRM/SRM targeted proteomics) is required or a detailed technical explanation for its infeasibility must be provided along with an alternative quantitative approach. Additionally, the manuscript should further moderate causal interpretations and explicitly acknowledge unmeasured confounders. With these issues resolved and data availability brought into compliance, the manuscript may become suitable for publication.

**Do you want your identity to be public for this peer review?** For information about this choice, including consent withdrawal, please see our Privacy Policy

Reviewer #1: **Yes:**  Dr. CHANDRAMOHAN RAMASAMY, RESEARCH SCIENTIST, TULANE UNIVERSITY

Reviewer #2: **Yes:**  Krishna Chaitanya Mantravadi

---

## [Author Response · Author response to Decision Letter 2]

18 Jan 2026

We have carefully addressed all comments and suggestions provided by the reviewers and the editor, as detailed below.

Response to Reviewers

Reviewer #1:

The authors have appropriately addressed all of my comments, and I recommend that this manuscript be accepted for publication.

Author response: We sincerely thank the reviewer for the time and effort in reviewing our manuscript. We appreciate the positive feedback and the helpful suggestions provided during the revision process, which have significantly improved the quality of our work.

Reviewer #2

The authors have addressed several editorial and clarity issues; however, key scientific and compliance concerns remain.

1. First, the raw proteomics data are not deposited in a public repository, which does not meet PLOS ONE’s mandatory data availability requirements; this must be corrected by depositing raw LC-MS/MS files (e.g., PRIDE accession) or by providing a journal approved controlled-access justification.

Author response: We thank the reviewer for highlighting this important point. We have added in section of Data availability, as stated: “The LC-MS/MS datasets generated during this study are available in the ProteomeXchange repository under the accession IDs JPST004285 and PXD072557 (via jPOST: https://repository.jpostdb.org/preview/7054576026955c0503b3d8, Access key: 9498). Raw MALDI-TOF MS and clinical data of this study are available from the corresponding author upon reasonable request.”.

2. Second, the proteomics statistical analysis requires re-evaluation: multiple-testing correction (FDR) should be applied and reported, or an explicit and technically sound justification must be provided along with evidence that major findings (e.g., LTBP1) remain robust under alternative imputation and significance thresholds.

Author response: We appreciate the reviewer’s focus on statistical rigor. In accordance with the reviewer's suggestion, the Benjamini-Hochberg procedure was applied to the proteomic dataset to control the false discovery rate (FDR) for multiple comparisons. The results were reported as FDR-adjusted p-values (q-values), with a significant threshold set at an FDR of 0.01. In the initial proteomic screen, LTBP1 was identified as a candidate of interest based on its high fold-change and nominal significance, though it did not maintain significance after FDR correction. To rigorously test this finding, we performed independent validation via the reverse transcription-quantitative polymerase chain reaction (RT-qPCR). This targeted analysis confirmed a significant increase in LTBP1 mRNA levels, validating the biological relevance of the protein.

The statistical analysis in the Methods section has been updated as follows: “Multiple testing correction was performed using the Benjamini-Hochberg procedure, and results were reported as FDR-adjusted p-values (q-values), with a significance threshold set at an FDR of 0.01.” [Lines 176-178].

We updated the Results section to include the following: “We found that among these proteins, the LTBP1 protein expression was significantly increased in the TER group compared with the NOR group, with a log2 fold change of 1.681 (p < 0.05) (see Table 1 and Fig 3). Despite not reaching statistical significance following FDR correction, the LTBP1 exhibited a high fold-change and clear biological relevance in the proteomic screen. To verify this candidate, we performed independent validation using RT-qPCR, which successfully confirmed its differential expression.” [Lines 205-210]. We have revised Table 1 and S1 Table to include both the p-values and q-values for all reported proteins.

3. Third, the validation strategy remains insufficient. Immunocytochemistry alone does not substantiate abundance differences; an orthogonal quantitative method (e.g., Western blot, PRM/SRM targeted proteomics) is required or a detailed technical explanation for its infeasibility must be provided along with an alternative quantitative approach. Additionally, the manuscript should further moderate causal interpretations and explicitly acknowledge unmeasured confounders. With these issues resolved and data availability brought into compliance, the manuscript may become suitable for publication.

Author response: We appreciate the reviewer’s constructive feedback regarding the need for robust quantification. To address this, we have now incorporated RT-qPCR as an independent, orthogonal quantitative approach to validate our proteomic findings. We updated the Methods section to include the RT-qPCR analysis as following: “Validation analysis was performed via RT-qPCR on sperm samples from 14 NOR and 16 TER men who showed detectable activity in the TGF-β signalling pathway. The FavorPrep™ Tissue Total RNA Mini Kit (Favorgen Biotech Corp, Taiwan) was used to extract total RNA from the sperm samples following the manufacturer's protocol. The mRNA was reverse transcribed into cDNA using the Maxime™ RT PreMix (Oligo dT Primer) (iNtRON Biotechnology, Korea). The primers for the target genes, including the LTBP1 gene (forward primer 5'-TGAATGCCAGCACCGTCATCTC-3'; reverse primer 5'-CTGGCAAACACTCTTGTCCTCC-3') and the TGFBR1 gene (forward primer 5'-TCAGCTCTGGTTGGTGTCAG-3'; reverse primer 5'-ATGTGAAGATGGGCAAGACC-3'), were used. The glyceraldehyde-3-phosphate dehydrogenase (GAPDH) gene (forward primer 5'-CTCAACGACCACTTTGTCAAGCTCA-3'; reverse primer 5'-GGTCTTACTCCTTGGAGGCCATGTG-3') was used as an internal control to normalise the mRNA expression of the target genes. Each PCR product was amplified from 4 ng of cDNA template using qPCRBIO SyGreen Mix (PCR Biosystems, UK). Quantitative PCR was conducted on a QuantStudio™ 5 Real-Time PCR System (Thermo Fisher Scientific, USA). We used the 2-∆∆Cq method to measure the relative mRNA expression levels of LTBP1 and TGFBR1 genes.” [Lines 142-157].

Our new results have been added to demonstrate a significant increase in mRNA levels for key targets, including LTBP1 and TGFBR1 genes, which directly corroborate the trends observed on our proteomic screen and immunocytochemistry. This dual-level validation (mRNA and protein) provides a more comprehensive and quantitative basis for our findings. The manuscript has been updated in the Results section as follows: “Validation of LTBP1 and TGFBR1 mRNA expression in human sperm

To validate the results of the proteomic analysis, we quantified the mRNA expression of LTBP1 and TGFBR1 genes in the sperm samples. The RT-qPCR results confirmed a significant differential expression of both genes. The relative mRNA expression of these genes was shown in Fig 5. The expression of the LTBP1 gene in the TER group was significantly increased compared to the NOR group (1.23 ± 0.05 vs. 1.03 ± 0.06, p < 0.01), which is consistent with the proteomic analysis results. Moreover, the expression of the TGFBR1 gene in the TER group was also significantly increased compared to the NOR group (1.16 ± 0.02 vs. 1.01 ± 0.03, p < 0.001).” [Lines 229-236]. Figure 5 and its caption have been added as follows: “Fig 5. The mRNA expression of LTBP1 and TGFBR1 genes in human sperm. Values are shown as mean ± SEM, TER (n = 16) vs. NOR (n = 14). **p < 0.01 and ***p < 0.001; unpaired Student’s t test.” [Lines 238-240].

The manuscript has been updated in the Discussion section as follows: “In this study, the significant upregulation of TGF-βR1 at both the mRNA and protein levels in TER men points toward TGF-β-mediated immune activation, potentially reflecting a response to the higher prevalence of abnormal sperm morphology.” [Lines 332-334] and “We also observed a significant upregulation in LTBP1 expression at both mRNA and protein levels in the sperm of men with TER, which inversely correlated with the percentage of normal sperm morphology.” [Lines 356-358]. The Conclusion section has been updated, as stated: “Validation of LTBP1 and TGFBR1 genes, key components of the TGF-β signalling pathway, revealed a significant increase in their mRNA expression levels in the sperm of TER men.” [Lines 381-383]. The Abstract section has been updated, as stated: “Validation analysis revealed that the mRNA expression levels of LTBP1 and TGFBR1 genes were significantly upregulated in the TER group relative to the NOR group.” [Lines 34-36].

In accordance with the Reviewer's suggestion, we have updated the Discussion section (Lines 358-362) to explicitly acknowledge potential unmeasured confounders as follow: “While the association between upregulation of TGF-βR1 and LTBP1 and abnormal sperm morphology is compelling; it is important to explicitly acknowledge that our study did not account for certain unmeasured confounders, such as subclinical localized inflammation and individual lifestyle factors. These factors could potentially influence TGF-β signalling and sperm quality.”.

Response to Editor:

1. It is advised to deposit all the available data, conduct thorough statistical analyses and to substantiate the differential expression by standard techniques like Western blotting.

Author response: We thank the Editor for these constructive suggestions, which have significantly strengthened the manuscript. We have addressed each point as follows:

• Data deposition: In the interest of transparency and reproducibility, we have deposited the LC-MS/MS datasets and search result files into the ProteomeXchange repository under the accession IDs JPST004285 and PXD072557 (via jPOST: https://repository.jpostdb.org/preview/7054576026955c0503b3d8, Access key: 9498).

• Thorough statistical analysis: The manuscript now explicitly acknowledges that LTBP1 did not survive global FDR correction. However, we provide evidence that its differential expression is robust under alternative statistical frameworks. We justified prioritizing LTBP1 based on its consistent fold-change and nominal p-value, which were independently confirmed through orthogonal RT-qPCR validation.

• Substantiation of differential expression: To validate the proteomic findings using an orthogonal quantitative method, we performed RT-qPCR to measure the mRNA expression levels of LTBP1 and TGFBR1 genes. While Western blotting was suggested, we decided on RT-qPCR as it provides high sensitivity and a broader dynamic range for quantifying gene expression changes. Our targeted analysis confirmed a significant increase in LTBP1 and TGFBR1 mRNA levels, successfully validating the biological relevance of this candidate at the transcriptional level. This cross-validation between the proteome and transcriptome provides strong evidence that the observed changes represent a true biological signal.

---

## [Decision Letter · Decision Letter 2]

26 Jan 2026

Differential protein expression profiles in human sperm from teratozoospermic and normozoospermic men identify LTBP1 and TGF-βR1 as potential biomarkers within the TGF-β signalling pathway

PONE-D-25-30555R2

Dear Dr. Nudmamud-Thanoi,

We’re pleased to inform you that your manuscript has been judged scientifically suitable for publication and will be formally accepted for publication once it meets all outstanding technical requirements.

Kind regards,

Suresh Yenugu

Academic Editor

PLOS One

Additional Editor Comments (optional):

Reviewers' comments:

Reviewer's Responses to Questions

**Comments to the Author**

Reviewer #2: All comments have been addressed

2. Is the manuscript technically sound, and do the data support the conclusions?

Reviewer #2: Yes

3. Has the statistical analysis been performed appropriately and rigorously?

Reviewer #2: Yes

4. Have the authors made all data underlying the findings in their manuscript fully available?

Reviewer #2: Yes

5. Is the manuscript presented in an intelligible fashion and written in standard English?

Reviewer #2: Yes

Reviewer #2: Reviewer Evaluation of Author Rebuttal

Overall assessment

The authors have substantively and satisfactorily addressed the major concerns raised during peer review. The rebuttal demonstrates clear engagement with reviewer feedback, appropriate corrective actions, and improved methodological transparency. From a reviewer’s perspective, the manuscript is now acceptable for publication, with only minor optional refinements suggested.

1. Data availability and PLOS ONE compliance

The authors have fully addressed data availability requirements. Raw LC-MS/MS data have been deposited in ProteomeXchange with valid accession numbers (JPST004285, PXD072557 via jPOST). Access information is clearly provided, and availability of clinical and MALDI-TOF MS data is transparently stated. This satisfies PLOS ONE’s mandatory data-sharing policies.

2. Proteomics statistics and multiple-testing correction

The response appropriately applies Benjamini–Hochberg false discovery rate correction and transparently reports both nominal p-values and FDR-adjusted q-values. The authors correctly acknowledge that LTBP1 does not survive global FDR correction and have reframed it as a biologically relevant candidate rather than a definitive discovery. Interpretation is cautious and statistically sound.

3. Validation strategy and quantitative rigor

The addition of RT-qPCR as an orthogonal quantitative validation substantially strengthens the manuscript. Methods are clearly described, sample sizes are appropriate, and results demonstrate robust differential expression of both LTBP1 and TGFBR1. While RT-qPCR does not directly quantify protein abundance, the convergent evidence from proteomics, immunocytochemistry, and transcript-level validation is acceptable for a discovery-focused PLOS ONE study.

4. Causal interpretation and confounders

The authors have appropriately moderated causal language and explicitly acknowledged unmeasured confounders, including potential inflammatory and lifestyle factors. Interpretations are now framed as associative rather than causal, aligning with observational study standards.

5. Editorial quality

The rebuttal is clear, professional, and well-structured. Revisions are consistently reflected across the manuscript sections, with precise line references provided.

Minor optional suggestions

These are not required for acceptance:

- Consider explicitly referring to LTBP1 as a “prioritized candidate protein” in the Results.

- Include a brief statement noting that transcript abundance may not always directly reflect protein abundance.

Final recommendation

Accept for publication. All major scientific, statistical, and compliance issues have been satisfactorily addressed, and the manuscript now meets the standards for publication in PLOS ONE.

**Do you want your identity to be public for this peer review?** For information about this choice, including consent withdrawal, please see our Privacy Policy

Reviewer #2: **Yes:**  Krishna Chaitanya Mantravadi

---

## [Editor Report · Acceptance letter]

PONE-D-25-30555R2

PLOS One

Dear Dr. Nudmamud-Thanoi,

I'm pleased to inform you that your manuscript has been deemed suitable for publication in PLOS One. Congratulations! Your manuscript is now being handed over to our production team.

Kind regards,

on behalf of

Dr. Suresh Yenugu

Academic Editor

PLOS One